# Disambiguating Symbolic Expressions in Informal Documents

**Dennis Müller**
Knowledge Representation and Management
FAU Erlangen-Nürnberg

Computational Logic
University of Innsbruck
d.mueller@kwarc.info

**Cezary Kaliszyk**
Computational Logic
University of Innsbruck

Institute of Computer science
Warsaw University
cezary.kaliszyk@uibk.ac.at

## Abstract

We propose the task of *disambiguating* symbolic expressions in informal STEM documents in the form of LaTeX files – that is, determining their precise semantics and abstract syntax tree – as a neural machine translation task. We discuss the distinct challenges involved and present a dataset with roughly 33,000 entries. We evaluated several baseline models on this dataset, which failed to yield even syntactically valid LaTeX before overfitting. Consequently, we describe a methodology using a *transformer* language model pre-trained on sources obtained from arxiv.org, which yields promising results despite the small size of the dataset. We evaluate our model using a plurality of dedicated techniques, taking the syntax and semantics of symbolic expressions into account.

## 1 Introduction

Despite huge advancements in machine learning, the task of understanding informal reasoning is still beyond current methods. In fact, it became commonplace that humans annotate informal documents containing reasoning in many domains, e.g. law (Libal & Steen, 2020). Reasoning is most visible in mathematical documents and software specification and as such in the last decades, the formalization of mathematical knowledge, and the verification of formal proofs, has become increasingly popular. By now, dozens of interactive and automated theorem prover systems are available, each providing libraries with up to hundreds of thousands of formalizations of mathematical definitions, theorems, and their proofs written by human mathematicians (Harrison et al., 2014).

While formal methods are still primarily used by computer scientists (e.g. to verify software and hardware, as well as in program synthesis), by now they have also drawn the interest of an increasing number of research mathematicians – primarily thanks to famous problems such as Kepler's conjecture (Hales et al., 2017) or the classification theorem for finite simple groups (Solomon, 1995), which have successfully been verified using theorem prover systems.

However, while *some* mathematicians have begun actively adapting formal methods for their work, there is a prohibitively large discrepancy between the way new mathematical results are developed, presented, and published in mathematical practice, and the way they are formalized and implemented in formal systems (Kaliszyk & Rabe, 2020): Most theorem proving systems implement a fixed *logical foundation* (such as variants of set theory or various kinds of type theories), a surface syntax in which a user declares new definitions and statements in terms of the underlying foundations, and either a tactic language or a language for expressing *proof terms* (usually on basis of the Curry-Howard-correspondence in a typed $\lambda$-calculus) that allow for declaring proofs. Consequently, the process of formalizing new content in a formal system resembles *programming* much more than it does developing informal proofs.

This discrepancy results in severe challenges for traditional mathematicians: Formal systems are difficult to learn and use, even if one is well acquainted with the (informal) mathematics involved. They require learning dedicated formal languages resembling programming languages, declaring content on a level of detail that is prohibitive for beginners even for "obvious" conclusions, and their

libraries are difficult to grasp without already being familiar with the system's language, conventions and functionalities. Due to the required level of detail, knowledge of the existing libraries is crucial when formalizing new content. Furthermore, many "intuitively valid" arguments can not be easily expressed in terms of a logical foundation in the first place, and knowing how to deal with those requires familiarity with the logical foundation involved and lots of practice.

Consequently, the utility of formalizing mathematical results can be too easily (and too often *is*) dismissed in light of the additional time and work required for non-experts. This is despite the fact that many services available for formal mathematics are already enabled by *semi*-formal (or *flexiformal*) representations, such as semantic annotations in natural language texts, or formal representations containing opaque informal expressions (see e.g. Kohlhase (2013); Lange (2011a); Iancu (2017); Kohlhase et al. (2017a); Corneli & Schubotz (2017); Dehaye et al. (2016)). Therefore, we need to invest into methods for bridging the gap between informal mathematical practice and (semi-)formal mathematics. One way to do so is to investigate *autoformalization*, the task of (semi-automatically) converting existing informal mathematical presentations to (increasingly) formal representations.

Notably, these issues extend beyond pure mathematics to other STEM (science, technology, engineering and math) fields, where the formal verification (or lack thereof) of results can have direct real-world implications – examples include an infamous and costly error in the floating-point unit of Intel processors (Harrison, 2003) and several human failures to adequately convert between SI and imperial units, most famously in NASA's Mars orbiter (Grossman). In fact, the former has already established formal verification as a vital tool in hardware design (Harrison, 2003).

Two observations motivate the research presented here:

1. The vast majority of STEM researchers can be assumed to be comfortable with using LaTeX; any integration of formal methods in a LaTeX development environment (e.g. via new packages or IDE integration) would consequently lower the entry barrier significantly.

2. The task of going from purely informal mathematical texts to fully formal representations of the contained knowledge is best done via a separation of concerns, by focussing on individual subtasks (such as disambiguating symbolic expressions, parsing natural language, and translating it to a formal foundation) using dedicated tools for each.

In this paper, we discuss specifically the task of *disambiguating* symbolic expressions – i.e. associating all symbols in an expression with their precise semantics – in LaTeX documents as a machine learning task, using sTeX semantically annotated LaTeX (Kohlhase, 2008). The contributions are threefold:

1. We discuss the details of disambiguating symbolic expressions in informal STEM documents as a neural machine translation task, 2. we present a new dataset specifically for this task, based on the existing SMGLoM library of sTeX macros (see Subsection 2.2), and 3. we present a methodology (using transformer language models) that allows us to achieve positive results on our dataset. We previously evaluated several baseline NMT models (such as Luong et al. (2017); Vaswani et al. (2017) and a plain character-based sequence-to-sequence model), which all failed to yield meaningful results due to our dataset being considerably smaller than is required for traditional NMT models.[1]

## 2 PRELIMINARIES

By *disambiguating*, we mean the task of transforming a sequence of symbols (representing a mathematical formula) into an *abstract syntax tree* and associating each leaf in the tree with a unique identifier specifying the precise semantics of the corresponding symbol.

While this might superficially seem an easy task, closer consideration shows that even obvious seeming statements such as "$a + b$" can in fact correspond to a multitude of possible disambiguations: $a$ and $b$ can be variables or previously defined constants, whereas $+$ can represent e.g. addition on multiple different number spaces, generic ring or vector space operations, or string concatenation. In order to adequately disambiguate expressions generically, it is, therefore, necessary to take the context in which the expression occurs into account.

---

[1]All code and data relevant to this paper is available at `https://gl.kwarc.info/dmueller/fifom`.

In this paper, we consider informal documents in LaTeX specifically, which we will disambiguate with the sTeX package, using semantic identifiers provided by the *SMGloM* library. This eventually enables various formal knowledge management services (such as type/proof checking) provided by the Mмт system.

## 2.1 sTeX

Kohlhase proposed sTeX (Kohlhase, 2008), a package for annotating LaTeX documents with structural and formal semantics which is today used by multiple groups formalizing mathematics in various systems. In particular, sTeX is based on OMDoc (Kohlhase, 2006), an extension of OpenMath (Buswell et al., 2004) which is foundation-agnostic in the sense that it does not favor a specific foundation (such as type or set theories) over any other. This approach is consequently best suited for semantifying informal documents, where foundations are often unspecified, left implicit or switched fluently. For example, category-theoretic and set-theoretic formulations are often used interchangeably in algebraic settings, whereas type theories are generally favored for computational aspects and formal systems.

Figure 1 shows example sTeX macros and their usage in various stages. Relevant for this paper is primarily the `\symdef` command, which introduces a new mathematical concept (e.g. `\nattimes` in Figure 1). It takes as arguments a macro name (e.g. `nattimes`), a symbolic notation (last argument) and optionally an OMDoc-name (e.g. `multiplication`), arity (e.g. `[1]`, which may be flexary) and notational precedence (e.g. `p=600`, for automatic bracketing). It generates a unique identifier for the concept being declared (based on the provided OMDoc-name), and a new LaTeX macro (e.g. `\nattimes`) for referring to the symbol. Alternative notational variants for symbols can be introduced via `\symvariant`, which are used as options to the macro (e.g. `\nattimes[cdot]`).

In addition to being valid LaTeX, compilable via `pdflatex`, sTeX-documents can be transformed to OMDoc using the LaTeXML-software (Ginev et al., 2011), yielding a formally disambiguated representation of the document and in particular the symbolic expressions therein on the basis of the macros provided by `\symdefs`. LaTeXML also heuristically attempts to disambiguate non-sTeX-symbols, e.g. by considering "=" and "+" as infix notations for generic equality and addition operators, respectively.

## 2.2 SMGLoM

The *SMGloM* (Kohlhase, 2014), *semantic multilingual glossary of mathematics*) is a library of hundreds of sTeX-modules containing mathematical concepts and definitions. It is separated into *signature modules* (using the `modsig`-environment, see Figure 1) containing only symbol declarations, and *natural language modules* (using the `mhmodnl`-environment, here exemplary for English) that serve as dictionary entries for these, in which the semantics of the symbols are described in a semi-formal manner. The second row of Figure 1 shows an SMGLoM entry.

## 2.3 Mмт

sTeX itself is integrated, and shares an underlying OMDoc ontology, with the Mмт system (Rabe & Kohlhase, 2013; Horozal et al., 2012; Rabe, 2017) – a foundation-independent meta-framework and API for knowledge management services. This integration makes the generic services provided by Mмт– e.g. type checking, library management/browsing, translation – available to informal mathematical texts. Using *alignments* (Müller, 2019; Müller et al., 2017), OMDoc-expressions can be translated between different libraries, languages and foundations. This allows for e.g. translating (originally) sTeX-content to a typed setting in order to e.g. check expressions and run type inference.

Additionally, several theorem prover libraries have been translated to OMDoc and integrated in the Mмт system, e.g. Kohlhase et al. (2017b); Müller et al. (2019) (for a detailed overview, see Müller (2019) and Kohlhase & Rabe (2020)). Extending these integrations to enable exporting from Mмт as well (and in conjunction with natural language processing), this could enable verifying informal mathematics imported via sTeX using external state-of-the-art theorem prover systems.

| sTeX declarations (signature module) | ```
\begin{modsig}{natarith}
...
 \symdef[name=multiplication]{nattimesOp}{\*}
 \symvariant{nattimesOp}{cdot}{\mathop\cdot}
 \symdef[assocarg=1,name=multiplication]
   {nattimes}[1]{\assoc[p=600]{\nattimesOp}{#1}}
 \symvariant{nattimes}[1]{cdot}
   {\assoc[p=600]{\nattimesOp[cdot]}{#1}}
... \end{modsig}
``` |
|---|---|
| sTeX references (natural language module) | ```
\begin{mhmodnl}{natarith}{en}
...
 \begin{definition}
  \Defi{multiplication} $\nattimesOp[cdot]$
  computes the \defi{product} $\nattimes[cdot]
    {a,b}$ (also written as $\nattimes{a,b}$ or
  $\nattimes[x]{a,b}$) of \trefiis[naturalnumbers]
    {natural}{number} $a$ and $b$. It is defined
  by the equations $\eq{\nattimes[cdot]{x,0},0}$
  and $\eq{\nattimes[cdot]{x,\natsucc{y}},
   \natplus{x,\nattimes[cdot]{x,y}}}$.
 \end{definition}
... \end{mhmodnl}
``` |
| PDF output (for the natural language module) | **Definition.** Multiplication $\cdot$ computes the product $a \cdot b$ (also written as $ab$ or $a \times b$) of natural numbers $a$ and $b$. It is defined by the equations $x \cdot 0 = 0$ and $x \cdot S(y) = x + x \cdot y$. |
| OMDoc | ```
<OMA>
  <OMS cd="smglom:mv?equal" name="equal"/>
  <OMA>
    <OMS cd="smglom:arithmetics?natarith"
      name="multiplication"/>
    <OMV name="x"/>
    <OMI>0</OMI>
  </OMA>
  <OMI>0</OMI>
</OMA>
``` |

Figure 1: An sTeX Example: The OMDoc corresponds to the symbolic expression $x \cdot 0 = 0$

## 3 STATE OF THE ART

Various papers over the last years have – explicitly or implicitly – attempted to extract formal information from informal documents using machine learning. These fall into two categories:

Firstly, there are projects that attempt to fully formalize informal mathematical documents using machine learning techniques, using the surface language of some theorem prover system directly as a target. In Kaliszyk et al. (2017a; 2015; 2014), the Flyspeck project (Hales et al., 2017) – the formalization of Kepler's theorem – was used as a basis for a parallel dataset in order to translate from informal mathematics to HOL Light (Harrison, 1996) syntax. Kaliszyk et al. (2017b); Wang et al. (2018; 2020) target the Mizar language (Mizar) instead, using the *Journal of Formalized Mathematics* (JFM) as data – an informal representation of the formal *Mizar Mathematical Library* (Bancerek et al., 2018).

While these projects achieved impressive results given the ambitious nature of the task, their success rate is naturally limited by the involved models having to solve several tasks at once (see second observation in Section 1), including ours. Additionally, by going to a fully formal language (and logical foundation) immediately, the result does not preserve the narrative presentation of the input document, effectively losing (for us) valuable information in the process. Consequently, our task and results obtained on it are not directly comparable to these projects.

Secondly, various projects have aimed to *solve* informally presented mathematical problems of various kinds. These include Arai et al. (2014); Matsuzaki et al. (2014; 2017; 2018) on pre-university math problems, Saxton et al. (2019) and Lample & Charton (2019) on high-school level equations,

Gan & Yu (2017) and Seo et al. (2015) on geometric problems, and Huang et al. (2018) and Wang et al. (2017) on solving typical high-school word problems.

While this naturally entails disambiguating symbolic expressions, all these projects reduce their domain of applicability to specific areas where all occurring formal symbols are syntactically unambiguous – primarily common arithmetic operations, functions, and relations on real numbers – such that disambiguation reduces to simple parsing of a fixed, small set of a priori known symbols.

## 4  TASK DEFINITION

**Definition 4.1.** *(Disamiguation Task) Let $\mathcal{L}$ be a set of LaTeX fragments (i.e. strings), which we assume are syntactically valid LaTeX in* some *suitable document context.*

*A* symbolic expression *is (for our purposes, simplified) any substring $s$ of some $S \in L$ such that $s$ is interpreted by the TeX-engine in* math mode *– e.g., if it is delimited by $, $$ or \[ and \] respectively.*

*For the purposes of our task, we call $S \in \mathcal{L}$* fully disambiguated*, if every symbolic expression occurring in $S$ only consists of:*

1. variable names *(e.g.* `n` *or* `\mathcal{G}`*, provided they do not represent specific, definite mathematical objects),*

2. *sTeX macros introduced via a* `\symdef` *declaration in the SMGLoM, or*

3. non-semantic *commands or characters, such as additional spaces/tabs/linebreaks, purely aesthetic spacing or kerning commands, unnecessary parentheses or clarifying comments (e.g. in under- or overbraces).*

*Let $\mathcal{L}_{sTeX} \subset \mathcal{L}$ the subset of fully disambiguated LaTeX fragments. Conversely, let $\mathcal{L}_{LaTeX} \subset \mathcal{L}$ be the set of LaTeX fragments that do not contain any sTeX macros[2].*

*Clearly, for any $S \in \mathcal{L}$, there is some $LaTeX(S) \subset \mathcal{L}_{LaTeX}$ such that $S$ and any $S' \in LaTeX(S)$ represent the same* symbolic presentation *– i.e. they generate the same output on* `pdflatex`*.*

*Conversely, we assume that for any $S \in \mathcal{L}$ there is a set $sTeX(S) \subset \mathcal{L}_{sTeX}$ such that 1. $LaTeX(S) = LaTeX(S')$ for all $S' \in sTeX(S)$ (i.e. they have the same symbolic presentation) and 2. all $S' \in sTeX(S)$ capture the* intended semantics *of $S$ - i.e. the author of $S$, were they to know the SMGLoM library sufficiently well, would agree that $S'$ is a correctly fully disambiguated variant of $S$.*

*Our goal is to learn a function $f : \mathcal{L} \to \mathcal{L}$ such that for any $S \in \mathcal{L}$ we have $f(S) \in sTeX(S)$.*

**Example 4.1.** *Consider the sentence from the SMGloM*

```
Multiplication $\cdot$ computes the product $a\cdot b$ (also written as
         $ab$ or $a\times b$) of natural numbers $a$ and $b$.
```

*The last two symbolic expressions (*`$a$` *and* `$b$`*) only consist of variable names, and are thus* considered fully disambiguated *already.*

*The first one (*`$\cdot$`*) refers to the multiplication operator on natural numbers, which in sTeX is represented as* `\nattimesOp`*, the remaining symbolic expressions are all multiplications on natural numbers applied to the variables $a$ and $b$ with different notations, represented in sTeX via* `\nattimes` *with various options.*

*We expect the target function $f$ on this input sentence to output*

```
Multiplication $\nattimesOp$ computes the product $\nattimes[cdot]{a,b}$
  (also written as $\nattimes{a,b}$ or $\nattimes[x]{a,b}$) of natural
                      numbers $a$ and $b$.
```

---

[2]Note that $\mathcal{L}_{LaTeX}$ and $\mathcal{L}_{sTeX}$ are not disjoint

## 5    DATASETS

We have two datasets of sTEX-content:

1. The SMGLoM[3], which introduces precisely those macros that we want to be learned by a model. Unfortunately, it provides relatively few symbols and hence can only cover a small part of informal documents even in theory. Additionally, apart from some rudimentary concepts such as logical connectives or basic arithmetic functions, the SMGLoM library *references* the majority of symbols only once (in the corresponding dictionary entry). This is unlike most other formal systems, where all symbols need to be typed or defined formally when being declared, which naturally leads to a significant number of references to previously declared symbols.

2. The MiKoMH[4]-repository of lecture notes by Michael Kohlhase (the author of sTEX) is heavily biased towards subjects in computer science, covering only a small part of SMGLoM-entries, and often introducing local `\symdef`s.

Notably, while the translation from source to target language is difficult, the *reverse* translation (from sTEX to plain LaTEX) is easy: Since sTEX macros internally expand (ultimately) to the plain notational representation as basic LaTEX, translating from the *target* to the *source* language amounts to merely expanding sTEX macros. This allows for easily generating a parallel dataset from a set of documents in the target language.

To obtain such a parallel corpus for supervised learning, we take the individual LaTEX-files in those repositories and do the following:

1. We separate the documents into small fragments of (on average) 500 character lengths, which we consider to be the *sentences* in $\mathcal{L}_{\text{sTEX}}$. Symbolic expressions occur preferably at the end of a sentence, based on the assumption that preceding text provides a more meaningful context for disambiguation. Sentences that do not contain symbolic expressions are ignored.

2. In each sentence $S = S_{\text{sTEX}} \in \mathcal{L}_{\text{sTEX}}$, we perform some standardization function which e.g. removes non-semantic macros and ensures that macro arguments are always braced, in order to minimize author bias,

3. We extract all symbolic expressions $(m_{\text{sTEX},i})_{i \leq n_S}$ in $S$ and expand all sTEX macros in them, resulting in $(m_{\text{LaTEX},i})_{i \leq n_S}$ (where $n_S$ is the number of symbolic expressions in $S$). Analogously, we expand all sTEX macros in $S$ itself, yielding $S_{\text{LaTEX}} \in \mathcal{L}_{\text{LaTEX}}$.

Each entry in our dataset then consists of a 4-tuple $(S_{\text{LaTEX}}, S_{\text{sTEX}}, (m_{\text{LaTEX},i})_{i \leq n_S}, (m_{\text{sTEX},i})_{i \leq n_S})$. In total, we obtain 911 entries from SMGLoM and 9200 entries from MiKoMH.

**Synthesizing Training Data**   In order to augment our datasets for supervised learning, we opted to exploit the MMT integration to synthesize additional training data.

For that, we aligned SMGLoM symbols with declarations in a strongly typed MMT archive; namely the *Math-in-the-Middle (MitM)* library (Müller, 2019). This allows us to randomly generate well-typed (and hence syntactically well-formed) terms in a typed setting, translate these along alignments to sTEX expressions and subsequently generate surrounding verbalizations.

The generating algorithm takes as input a set of symbols Sym (e.g. all MitM-symbols for which an alignment to SMGLoM exists) and a starting symbol $s \in$ Sym (e.g. `nattimes`; binary multiplication on natural numbers). It returns a random well-typed formal expression $t$ which is guaranteed to contain $s$. Afterwards, it is *verbalized* as an sTEX sentence using natural language fragments (a detailed description of the algorithm is given in Appendix A).

The synthesized sTEX sentences are then treated as above to augment our parallel training corpus.

As an **evaluation dataset**, we developed sTEX documents based on selected fragments of introductory sections from mathematics lecture notes; primarily containing basics such as set operations, number

---

[3]`https://gl.mathhub.info/smglom`
[4]`https://gl.mathhub.info/MiKoMH`

spaces, examples for proofs by induction, basic combinatorics, and definitions of common algebraic structures, containing 161 symbolic expressions in total. Importantly, these documents were written by hand, with a focus on featuring multiple symbols with the same symbolic representation; primarily the usual arithmetic operations on different number spaces.

Of the $\approx 100$ SMGLoM symbols used therein, 92 were aligned with corresponding symbols in the MitM library and used as input symbols for synthesizing sentences; with 250 sentences per starting symbol (as to not drown out the non-synthesized sentences), yielding 23,000 additional sentences.

Unlike the training datasets, the evaluation document was translated to plain LaTeX manually using the PDF as a reference, in order to avoid possible spurious patterns in automatically expanded sTeX.

## 6    sTeX-Annotating with Machine Learning as an NMT Task

In the course of our experiments, we considered our disambiguation task as a machine translation (NMT) problem, the models for which have been proven to be quite effective even beyond natural language translations (Clark et al., 2020). In fact, the autoformalization projects mentiond in Section 3, which are spiritually closest to our task, all used NMT models with positive results. There are however several aspects that distinguish a LaTeX-to-sTeX translation from similar translation tasks which significantly affect the applicability of existing tools and hence our methodology.

First, Unlike the most popular formal systems, there is no large library of formalizations for the translation target. This leaves us with only a small dataset that (for the reasons outlined in Section 5) does not represent well the general distribution we would like to learn.

Second, translation is only relevant for specific fragments of an input text, namely the symbolic expressions; for the surrounding natural language texts, translation should be the identity. Nevertheless, surrounding text usually contains critical information for disambiguation; e.g. without the surrounding context, it is impossible to disambiguate an expression $a + b$, since the symbol "+" could refer to any of dozens of addition operations.

Finally, depending on perspective, the domain language is a proper subset of the target language; or rather (since we want to avoid ambiguous expressions in sTeX) domain and target language share both a basic grammar as well as a large amount of vocabulary (namely $\mathcal{L}_{\text{LaTeX}} \cap \mathcal{L}_{\text{sTeX}}$) which e.g. subsumes natural English. For the domain language, large datasets are easily obtainable.

Our task could also be considered as a *text style transfer* task – e.g. Yang et al. (2019) uses pre-trained language models for text style transfer, roughly similar to (but more sophisticated than) our approach. While the datasets used therein are still considerably larger than ours, this might be a promising avenue for future improvements over our model.

## 7    Methodology

Notably, sTeX macros reflect the *syntax tree* of an expression, so that on symbolic expressions *alone*, the representation of the target sequences is naturally analogous to those chosen in *string-to-tree* translations (Aharoni & Goldberg, 2017). Plain LaTeX however is not naturally amenable to a tree-structured representation, making *tree-to-tree* approaches (Chen et al., 2018) not easily applicable to our dataset.

Initial experiments using standard, dedicated NMT models with full sentences as input/output quickly proved to be ineffective due to the size of the training corpus, which was too small to cause these models to even generate syntactically correct LaTeX (e.g. knowing to balance pairs of brackets) before overfitting on the training data. This makes it difficult to compare our approach to an informative baseline model.

*Transformer language models* (e.g. Devlin et al. (2018); Liu et al. (2019); Radford (2018); Radford et al. (2019); Clark et al. (2020)) allow us to leverage huge available corpora of plain LaTeX documents to train a model to "understand" both basic LaTeX syntax and mathematical terminology. Using those, we consequently do not need to rely on our small dataset for this base-level understanding. We can then approach learning sTeX annotations as a downstream task on a pre-trained transformer model. Consequently, we pre-trained a GPT2 (Radford et al., 2019) model on a large portion of available

LATEX sources of scientific papers from the preprint repository `arxiv.org` (6,673,950 entries of length 1,024 tokens). The model was trained *from scratch* in order to use a dedicated tokenizer trained on LATEX directly (byte-level tokenizer; vocabulary size 32,000) rather than natural language alone.

In order to leverage the pretrained model for both source and target language[5], we subsequently opted to fine-tune the GPT2-model on inputs of the form

$$S_{\text{LATEX}} \texttt{  } m_{\text{LATEX}} \texttt{  } m_{s\text{TEX}} \texttt{ },$$

where `` a single-token separator.[6] For example, for Figure 1 the training data contains fragments (normalized) such as:

```
Multiplication $\cdot$ computes the product $a\cdot b$ (also written as
        $ab$ or $a\times b$) of natural numbers $a$ and $b$.
             $a\cdot b$  $\nattimes[cdot]{a,b}$ 
```

We then use text generation on inputs of the form $S_{\text{LATEX}} \texttt{  } m_{\text{LATEX}} \texttt{ }$ for translating and stop generating after encountering ``.

By using one entry per symbolic expression, we obtain a dataset of 121,368 examples. The GPT2-model was finetuned on these for five epochs, resulting in an average training loss of 0.04 and yielding promising results on the evaluation set (see below). This approach has the following advantages:

1. It allows for using large datasets of generic LATEX documents to learn basic syntactic rules and semantics of mathematical expressions beyond our small sTEX datasets.

2. We conjecture that this approach makes the model less sensitive to spurious patterns in the synthesized part of our dataset.

3. Adding new symbols to the SMGLoM and aligning them to (new or existent) symbols in the MitM library allows for immediately synthesizing training data, obviating the need to first obtain large amounts of data *using* the new symbol before the model can learn to use it.

4. The mere pretrained GPT2 model can be trained on *additional* downstream tasks, e.g. introducing macros for referencing mathematical concepts in natural language fragments.

## 8 EVALUATION AND RESULTS

The traditional evaluation metrics (loss during evaluation, perplexity, BLEU) are somewhat difficult and/or meaningless to apply in our situation, since 1. the returned tokens and provided label tokens might differ in semantically irrelevant ways (e.g. `$a+b$` vs. `$a + b$`), and 2. loss/perplexity would be evaluated during a forward pass in a next token prediction task on a token-by-token basis, which would retroactively "correct" errors in prediction that would otherwise yield completely wrong result.

Consequently, we opted for a plurality of evaluation strategies. Let $S_F$ the returned sentence of our model on an input $S_{\text{LATEX}}$ with the correct label $S_{s\text{TEX}}$. Then on our evaluation set we get

1. $S_F \in \mathcal{L}$ for 96.9% of inputs

2. $S_{\text{LATEX}} \in \text{LATEX}(S_F)$ for 64.0% of inputs,

3. $S_F \in \mathcal{L}_{s\text{TEX}}$ for 60.2% of inputs, and

4. $S_F = S_{s\text{TEX}}$ for 47.2% of inputs.

In comparison, using traditional NMT models auch as Luong et al. (2017); Vaswani et al. (2017) we effectively obtained 0% success rates for all of the above. Additional evaluation techniques exploiting the MMT integration are described in Appendix B.

Figure 2 shows a few examples where our model "failed" in interesting ways. As the first and fourth examples show, the model seems to consistently fail to replace "=" by the intended macro $\eq - a$ failure that LaTeXML can recover when converting to OMDOC, but also regularly occurs in the training data. Similarly, $\ldots$ often leads to wrong translations: The first example shows that the

---

[5]Initial experiment with the pretrained model as encoder component only showed improvements over randomly initialized encoder-decoder-models, but ultimately proved unsuitable still due to the small dataset size.

[6]inspired by `http://jalammar.github.io/illustrated-gpt2/` `#part-3-beyond-language-modeling`

```
S_LATEX:   \mathbb{N}=\{0,1,2,3,\ldots\}
S_sTEX:    \eq{\NaturalNumbers,\setdots{0,1,2,3}}
S_F:       \NaturalNumbers=\set{0,1,2,3}
─────────────────────────────────────────────────────────────
S_LATEX:   (A \subseteq B)\Leftrightarrow(\forall x\in A. x\in B)
S_sTEX:    \biimpl{\sseteq{A}{B}}{\foral{\inset{x}{A}}{\inset{x}{B}}}
S_F:       \biimpl{\sseteq{A}{B}}{\foral{x}{A}\inset{x}{B}}}
─────────────────────────────────────────────────────────────
S_LATEX:   \mathcal{P}(A):=\{x|x\subseteq A\}
S_sTEX:    \defeq{\powerset{A}}{\setst{x}{\sseteq{x}{A}}}
S_F:       \defeq{\powerset{A}}{\bsetst{x}{x}{\sset{x}{x} A}}
─────────────────────────────────────────────────────────────
S_LATEX:   1+2+3+4+5=(5\cdot6)/2=15
S_sTEX:    \eq{\natplus{1,2,3,4,5},\natdiv[slash]{\nattimes[cdot]
              {5,6}}{2},15}
S_F:       \natplus{1,2,3,4,5}=\natdiv[slash]{\natplus{\nattimes[cdot]
              {5,6},4,5}}{2}=15
```

Figure 2: Example Inputs and Outputs from our Evaluation Set

model simply dropped `\ldots`, using a generic set constructor macro `\set` rather than `\setdots`, the one specifically intended for sets ending in ellipses.

In the second example, the model seems to introduce a nonsensical additional argument for the `\foral` macro. Notably, the expression $\forall x \in A.P$ can also be achieved using the dedicated macro `\foralS{x}{A}{P}`. Seemingly, the model chose the macro `\foral`, and the arguments for the `\foralS` macro, yielding a wrong translation that generates a wrong pdf output, while being "semantically almost correct".

In the third example, the model confuses the macro `\setst` (for set comprehension) with a more complex macro `\bsetst` (for set comprehension with a complex pattern on the left side). Additionally, it confuses `\sseteq` (for inclusive subsets $x \subseteq A$) with `\sset` (for generic subsets $x \subset A$), duplicating the first argument and moving the *intended* argument `A` outside the scope of the macro.

Example four is interesting in that the model correctly identifies the arithmetic operations as those on the natural numbers, but spuriously inserts an additive term `\natplus{...,4,5}`; this is likely an artifact from the left-hand side of the equation. Interestingly, these kinds of artifacts occur more than once in our evaluation set.

## 9 CONCLUSION

We have proposed the task of disambiguating symbolic expressions in informal STEM documents and defined this task formally. This allows for annotating informal documents semantically, and further processing them using tools that support such annotated documents (e.g. MMT). We discussed the specificity of this task and what separates this task from other NMT problems. We developed a dataset for this task and presented an approach that yields promising results, especially in light of the size of the dataset. In particular, the presented approach points to the efficacy of using transformer models pretrained on generic LATEX documents.

In the future, we plan to combine the proposed symbolic disambiguation approach with an auto-formalization framework. This way we aim to achieve better results for end-to-end formalization of informal mathematical documents. Furthermore, more promising results for the currently proposed task could be obtained by reintegrating the proposed models into an encoder-decoder NMT model.

ACKNOWLEDGMENTS

The first author and this work were supported by a postdoc fellowship of the German Academic Exchange Service (DAAD).

The second author is supported by ERC starting grant no. 714034 *SMART*

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

## A  SYNTHESIZING TRAINING DATA

The generating algorithm takes as input a set of symbols Sym (e.g. all MitM-symbols for which an alignment to SMGLoM exists) and a starting symbol $s \in$ Sym (e.g. nattimes; binary multiplication on natural numbers). The algorithm then proceeds as follows:

1. If $s : T$ has a (simple or dependent) function type, we fill in the required arguments. For $s =$nattimes, our type is $T =$Nat→Nat→Nat, hence we need to find two arguments $s_1, s_2$ of type Nat. For each $s_i$ of required type $T_i$ we proceed as follows:

   (a) With probability $p_{var}$, we introduce a new variable $v : T_i$ from a list of allowed variable names (which include variants such as $a$, $a'$, $a_0$ etc.) and let $s_i := v$.

   (b) With probability $p_{fun}$, we pick a symbol $f \in$ Sym with a function type with return type $T_i$ (e.g. for $T_i =$Nat, we can pick natplus). In that case, we let $s := f$, recurse, and set $s_i$ as the result.

   (c) With probability $p_{const} = 1 - p_{var} - p_{fun}$, we pick a constant symbol $c \in$ Sym of type $T_i$ (e.g. for $T_i =$Nat we can pick 0) and return $s_i := c$.

   In order to avoid stack overflows, we reduce $p_{fun}$ in each iteration by a certain factor $< 1$. As to not overuse certain symbols, we scale $p_{fun}$ and $p_{const}$ with the number of respectively suitable symbols available; if Sym contains no suitable function or constant symbols, we let $p_{fun} = 0$ (and/or $p_{const} = 0$, respectively).

2. If $s : T$ does *not* have a function type (or all its parameters have been filled in 1.), then $s$ is well-typed and we return $s$ with probability $1 - p_{up}$.

   With probability $p_{up}$, we instead pick a new symbol $s_f \in S$ of some function type such that some $i$-th parameter type of $s_f$ is $T$. In that case, we let $s_i := s$ and $s := s_f$ and recurse.

   Again, in order to avoid stack overflows we reduce $p_{up}$ by some factor with each iteration.

The algorithm also takes subtyping into account, e.g. whenever a term of type Real is required, terms of type Int or Nat are used with some probability.

In order to obtain a sentence in the sense of Section 5 providing context for disambiguation, we first translate $t$ along alignments to SMGLoM (using a random \symvariant), collect the set $V$ of all free variables of $t$ and *verbalize* their types. For that, we associate each *type* with a set of *verbalizations* from which we choose randomly to produce a sentence that introduces the variables before using them in the generated expression. Figure 3 shows a few example verbalizations for a variable $x$ of type Nat and generated sentences for the input symbol $s =$realuminus; the negation on real numbers.

The verbalizations are categorized as *prefixed* (e.g. "*a natural number $n$*") or *suffixed* (e.g. "*$n$ a natural number*"), and *singular* or *plural*, and picked according to the number of variables of the same type and the surrounding sentence, which is also picked at random (e.g. "*Assume we have ...*" uses prefixed, whereas "*Let ...*" uses suffixed).

## B  EVALUATION TACTICS

For every LATEX input $S_{\text{LATEX}}$, expected label $S_{\text{sTEX}}$ and returned sentence $S_R$, we employ the following strategies, the results of which are summarized in Figure 4:

islatex  We parse $S_R$ into an AST. Success implies that $S_R$ is syntactically valid LATEX. This might fail for "minor" reasons such as a missing closing bracket. It might yield false positives in cases where macros (not explicitly considered by our parser) occurring in $S_R$ have a wrong number of arguments.

All subsequent evaluation strategies require islatex to succeed.

stexcheck  We heuristically check whether $S_R$ is in $\mathcal{L}_{\text{sTEX}}$ – unlike islatex, this requires that all sTEX macros occurring in $S_R$ have the right number of arguments. Success does *not* tell us that the input has been disambiguated *correctly*, but *does* imply that is *has* been disambiguated *at all*. False negatives can occur if $S_R$ (and thus likely $S_{\text{LATEX}}$ as well)

|  | Generated sTeX | PDF output |
|---|---|---|
| Verbalizations | `$\inset{x}{\NaturalNumbers}$` | $x \in \mathbb{N}$ |
|  | a positive integer `$x$` | a positive integer $x$ |
|  | an integer `$\intmethan{x}{0}$` | an integer $x \geq 0$ |
|  | a natural number `$x$` | a natural number $x$ |
| Sentences | `Assume we have some $\inset{y'}{\NaturalNumbers}$ and arbitrary $\inset{\mathcal F}{\IntegerNumbers}$. It follows that $\realuminus{\realuminus{\inttimes[x]{\mathcal F,y',y'}}}$.` | Assume we have some $y' \in \mathbb{N}$ and arbitrary $\mathcal{F} \in \mathbb{Z}$. It follows that $--(\mathcal{F} \times y' \times y')$. |
|  | `Let $\natmorethan n{0}$. Then consider $\realuminus{\realuminus{\natsucc{\natsucc n}}}$.` | Let $n > 0$. Then consider $--S(S(n))$. |
|  | `Whenever we have some positive natural number $\varepsilon$, any integer $\ell$ and a real number $\livar{\mathcal C}{2}$, then it follows that $\realtimes{\livar{\mathcal C}{2},\livar{\mathcal C}{2},\realplus{\realuminus{\ell},\natsucc{\varepsilon}}}$.` | Whenever we have some positive natural number $\varepsilon$, any integer $\ell$ and a real number $\mathcal{C}_2$, then it follows that $\mathcal{C}_2\mathcal{C}_2(-\ell + S(\varepsilon))$. |

Figure 3: Example Verbalizations for $x$ :Nat and Generated Sentences

contains complex variable names, or if $S_R$ contains e.g. an equality symbol "=" instead of the corresponding sTeX macro, which LaTeXML could recover.

`eval_latex` All sTeX macros occurring in $S_R$ are expanded and $S_R$ is normalized as described in Section 5. The result is string-compared to $S_{\text{LaTeX}}$. Success thus implies, that the notational presentation in PDF output of $S_{\text{LaTeX}}$ and $S_R$ will coincide. False negatives can occur due to minor differences e.g. in not strictly necessary brackets.

`omdoc` $S_R$ is translated to OMDoc using LaTeXML and imported to MMT. Success guarantees syntactic well-formedness of $S_R$. Since both the LaTeXML-OMDoc export and the subsequent MMT-import are somewhat brittle, this can easily lead to false negatives.

`translated` The import from `omdoc` is translated to the typed MitM library. This entails that all symbols used in $S_R$ are aligned with MitM symbols and $S_R$ is amenable for formal knowledge management services.

`inferred` The translation to MitM obtained from `translated` is type checked by MMT by having its type inferred. Success guarantees that $S_R$ is well-typed.

Notably, if $S_R$ is a mere variable (e.g. the expression `$n$`), it does not actually have an inferrable type, but succeeds trivially. This accounts for 60 of the entries in our evaluation set, i.e. 37%.

`provided_stex` Both the expected label $S_{\text{sTeX}}$ and $S_R$ are normalized and string-compared. Success implies that $S_R$ is definitely the correct translation. False negatives can easily occur due to non-semantic differences between $S_{\text{sTeX}}$ and $S_R$ however, such as bracketing, nested applications in $S_R$ (e.g. `$\natplus{\natplus{a,b},c}$` vs. `$\natplus{a,b,c}$`), etc.

`stex_as_omdoc` $S_{\text{sTeX}}$ is translated to OMDoc via LaTeXML and directly compared to the OMDoc-term obtained from `omdoc`. Like `provided_stex`, success implies that $S_R$ is correct, but it is more fault-tolerant with respect to the precise syntax of $S_R$, while being *less* fault tolerant due to the issues mentioned in `omdoc`.

The first three evaluations can always be applied; from the remaining, all but `provided_stex` require a working installation of LaTeXML and its sTeX-Plugin. The last two require a known correct translation.

| Total inputs | 161 |
|---:|:---|
| islatex | 96.9% |
| stexcheck | 60.2% |
| eval_latex | 64.0 % |
| omdoc | 76.4% |
| translated | 63.5% |
| inferred | 59.6% |
| provided_stex | 47.2 % |
| stex_as_omdoc | 53.4 % |

Figure 4: Results on our Evaluation Document

A detailed log file on our evaluation document with the individual results for each input and evaluation is available in the associated git repository.

