# OpenReview forum: "Disambiguating Symbolic Expressions in Informal Documents"
_ICLR.cc/2021/Conference — ICLR 2021 Poster_

### Official Review · AnonReviewer2 · 2020-10-23
**A thorough paper about style transfer: From Latex math expressions to less formal descriptions. A thorough work, but the problem definition can be clearer and connection to previous work can be better made.**

**Rating:** 7
**Confidence:** 4

**Review:**

This paper addresses a variant of the style transfer problem - that is, transferring formal latex expressions to less formal descriptions that can be followed by mathematician.

This is a paper that formulates a new task, provides a dataset for the task and tests initial approaches for solving it. It is important to take this paper type into account when reviewing it  - I am not expecting a very creative solution, or very strong results at this stage. What is important to me when reviewing such a paper is to see an accurate and interesting problem definition, an appropriate dataset, modeling and experiments that demonstrate the challenge of problem and of the evaluation (if evaluation is indeed challenging) and proper awareness of previous work.

It will be most straight forward for me to review the paper by listing its pros and cons.

Strong points:

1. The problem exposition is through and clear, and the introduction surely provides good motivation for the problem.

2. It is clear that the authors are expert on the subject matter. That is, they are deeply familiar with the problem and with directly related previous word (that is, previous work that addressed this very problem or close variants).

3. The authors propose a new dataset that is likely to be useful for the community of researchers that work on this problem.

4. The paper proposes an algorithmic approach for the problem, tests it in experiments with the new dataset and the authors are aware of potential challenges in the evaluation and try to address them.

Weak points:

1. The problem definition was not clear to me. I surely understand the general idea but I am missing a concrete example that demonstrates what exactly an algorithm for the problem gets as input and what is its output.

2. I had a similar problem with the description of the dataset. Yes, there is a formal description (just as there is a formal description of the task), but the lack of examples leaves the description at a very abstract level - I could not understand what exactly should be expected in the dataset.

I should note that 1+2 makes it harder to evaluate the results and to evaluate the appropriateness of the evaluation.

3. The authors does not show awareness of work in semantic parsing and in style transfer.  These works are very important both for the algorithmic approach and for understanding the challenges of evaluation (e.g. plurality).  For example, there is lots of semantic parsing research on transferring text into SQL queries (the opposite direction of the current problem) or on solving textual mathematical problems. Can any ideas be borrowed from this literature or is it only vanilla MT that can be applied ? As said above, I am aware this is a paper that introduces a task, but part of this introduction should be, I believe, the connection to relevant ideas and approaches.

Overall, despite the challenges, I think the paper can contribute an interesting and practical new task to the research community - both at the task definition level and in providing an actual dataset. I recommend that the authors try to solve the above issues in the final version, but I am leaning towards acceptance.

---

> ### Author Response · Authors · 2020-11-16
> **Our reply**
>
> You seem to be mostly interested in a better presentation and examples for the precise input/output of the model. One such example can be found in Section 6 (on page 7) - could you be more specific in what you want to see? Given that in the final version we have an extra page, we can happily add more informative examples.
>
> > For example, there is lots of semantic parsing research on transferring text into SQL queries (the opposite direction of the current problem) or on solving textual mathematical problems. Can any ideas be borrowed from this literature or is it only vanilla MT that can be applied ?
>
> As far as we know, the projects mentioned use purely "classical" algorithmic approaches for (semantic) parsing, because in those settings, these are sufficient. For example, the projects on solving mathematical problems (as far as we know) all focus on specific mathematical domains (e.g. real arithmetics), where the meaning of symbolic expressions is (assumed to be) unambiguous anyway or can be easily inferred from purely syntactic usage (e.g. "+" is always assumed to be addition on real numbers), so that the expressions can simply be parsed deterministically. A paragraph to that effect can be found at the bottom of the state of the art section (page 4).

---

### Official Review · AnonReviewer3 · 2020-10-30
**Important line of work hindered by little methodological novelty and poor evaluation**

**Rating:** 4
**Confidence:** 3

**Review:**

#### Summary:

In more mathematical fields, theorem provers and similar systems can validate claims made about formal systems.  However, many research contributions come in the form of papers, and thus they are never validated in this way.  Math researchers can express their contributions in a special purpose language to do this, but that places an additional burden on them to learn this skill.

An alternative would be to "translate" the research into formal languages which could be operated on by automated systems.  This seems to be the goal of this paper, which looks at using translation to disambiguate some expressions from STEM documents written in LaTeX, which maps it into an sTex document.  Previous research in this area used more hand-specified transformations, and was evaluated on different data.  This makes this work closely related to existing work, but not directly comparable.

The main finding of this work is that pre-trained transformer models outperform more traditional fully-supervised translation systems on this task.  It is difficult to guage precisely to what extent the proposed method solves the task, or to fully grasp what aspect of the translation problem is being solved.



#### Strong points:

The proposed approach of using transformers and large in-domain pre-training is similar to a lot of recent work which has shown to work well in practice, and is therefore well-motivated.  The task itself is important and improvements in this area could have broad range of impact that would even be good to the ML field itself, so even a practical improvement using fairly standard ML would be a useful contribution.

The authors are clearly knowledgable on the topic, and discuss and cite a great deal of the literature and libraries relevant to the problem.  It's a heavy paper -- there's a very extensive set of work that's been studied and referenced.


#### Weak Points

In the start of section 4, a number of systems were listed.  Each of these was an attempt to automate the formalization process, but there was no attempt to compare against these methods.

As noted towards the end of section 4, the target transformation for much of the document is the identity.  Given that actually a lot of this text should not change, is phrasing it as translation the best choice?  It seems that phrasing it this way sets up the NMT baselines to perform poorly, since a lot of training is necessary to prime the model to learn this identity transformation, and that data is not given to those models.

The evaluation methodology is confusing.  For instance, it seems some of the data is generated via an automated procedure, both in the supervised learning (Section 5) and in the Synthesizing Training Data section.  It makes it difficult as a reader to understand why this is not a chicken-and-egg type of scenario: if automated methods produce the dataset, which is then used to train the model, then why are those methods not sufficient for the end task?  This may be a problem that arises from introducing so many domain-specific libraries and formalisims, that it leaves the reader with a great deal of difficult to understand precisely what the transformation is accomplishing and from what type of data.

It was also bizarre in the results section how the baselines were dismissed in writing, their results were never presented.  If the baselines are truly that bad, then do they suffice as baselines?  The authors choose these instead of the existing formalization methods, so why make the contrast with methods that are not in a position to perform the task well?


#### Recommendation

Because the presentation makes it difficult to fully grasp the problem setting, precisely what is being learned, precisely what is failing, it is difficult to recommend the paper for acceptance.  It is actually very understandable that this particular paper has this problem, because the authors are forced to introduce many unfamiliar concepts -- the problem setting, the types of formalisms used, the libraries used in creating the data, etc.  These are all things that are outside the scope of the typical ICLR paper and thus warrant a clear introduction, but space is limited.  I could easily imagine this paper filling up 12-14 pages just with the same content presented here.  But ultimately the paper is not written in a way that can properly convey the scope of the work and narrow in on precisely the targetted problem and why it's difficult and important.

Then the experimental section is quite short and lacks important comparisons.  Given the lack of suitable baselines, I would not be able to recommend accepting the paper without real comparisons to other work in this area.  Again this could be a space concern, but the paper overall spends too much time leading up to methodology/experiments, and then is very light on actual experimental content.  Factor in that the model is used in a very off-the-shelf way, and doesn't treat the problem setting really any different than a standard translation task, it is hard to see real novelty in the modeling contribution either.

Overall I think the work is promising, but it is far too rough in its current state to be considered for acceptance without significant revision.  It would need major restructing and refocusing, more experiments, and more analysis.


#### Presentation

I feel like there's a lot of domain specific meanings to terminology that makes it more difficult than necessary to understand by a general ML audience.  Take for instance, formal and informal.  To most language users, a scientific paper is a formal document -- it uses formal language.  So it takes me some time as a reader to get into the actual data section and understand truly what is meant by informal here.  There are many things of this nature that would be better to clarify up-front, so the reader with the typical ML background and biases doesn't carry around incorrect concepts of what the paper is about, for longer than is necessary.


The citation format is incorrect.

Small typos throughout.


#### In considering author response:

Thank you to the authors for continuing discussion on the points raised in my review, and for further clarifying the nature of the data as a kind of unidirectional ambiguity problem.   I understand this better now and can see a contribution in releasing this data / data-generating process for other researchers studying autoformalization.  On account of this I'm going to raise my initial scoring.

On the subject of methodology, I still think there are reasons to reconsider this work.  As discussed, the translation baselines were not great.  I think it's not really fair to compare those models without pre-training on data that was too small to learn basic tree properties.  It is possible that translation models that perform string-to-tree translation would perform better here(1), though results from natural language translation would hint towards the pre-trained models still performing better.  Translation models used in the domain of programs seem more suitable as well, and there's a good number of these, and there is a natural desire to generate strings that reflect a properly nested tree (2).  There is also work on mapping strings to knowledgebase queries that seems similar in input/output (IIRC, Luke Zettlemoyer's had a number of important papers in this line).

But at best these would still be comparisons of mostly off-the-shelf translation models, which doesn't leave the reader with much of a takeaway.

So I'm left feeling that if the authors want a useful quantitative comparison, these methods should be explored.  Pre-trained model beats model trained on only in-domain data is not to me a story significant enough to warrant inclusion in the conference, even if it contributes a new dataset (as the modeling is presented as a contribution here).  Even off-the-shelf methods can of course be part of an important contribution when the authors show that they have pushed the field further with an important result (say, GPT) but I do not feel the evaluation in this case supports that conclusion.

It seems more natural given that none of these methods are likely to out-perform a vanilla pre-trained translation model, that the problem description and qualitative evaluation are of the utmost importance.  I would really recommend expanding this beyond half a page, to give the reader a better idea of what problems are solved and what are remaining.  It also seems that some of the errors pointed out (like those involving ellipses) would likely be remedied by additional synthetic data.  As I'm the most dissenting reviewer, I would still hope the authors attempt to improve the results section with the additional page upon acceptance.

1.
Towards String-To-Tree Neural Machine Translation
Roee Aharoni, Yoav Goldberg

2.
Tree-to-tree Neural Networks for Program
Translation
Xinyun Chen, Chang Liu, Dawn Song

---

> ### Author Response · Authors · 2020-11-16
> **Our reply**
>
> > In the start of section 4, a number of systems were listed. Each of these was an attempt to automate the formalization process, but there was no attempt to compare against these methods.
>
> A direct comparison of our approaches with autoformalization is not possible, as autoformalization is a translation to a fixed logic, whereas our translation still allows the logic used in the target language to be arbitrary. As such the tasks are different. In fact autoformalization is useful for translation to the languages of proof assistant systems, whereas what we do here is useful for computer algebra systems and similar mathematical knowledge management and interchange tools. In particular (as mentioned in the state of the art section, page 4), we deliberately do not want to change the presentation of informal natural language fragments, which those other projects translate to logical statements in a formal language.
> Consequently, a comparison between ours and autoformalization projects would require both a shared evaluation dataset (see our reply to AnonReviewer1 regarding the difficulties there), and aligning the purely symbolic expressions in the informal inputs with their corresponding counterparts in the translated fully formal outputs of the autoformalization models - which requires an extreme amount of manual work and expertise in both the mathematics involved as well as the formal system we compare our approach to. Consequently such a comparison is currently not feasible.
>
> > As noted towards the end of section 4, the target transformation for much of the document is the identity. Given that actually a lot of this text should not change, is phrasing it as translation the best choice?
>
> It seems to us that machine translation is the established method that comes closest to what we are trying to accomplish, and all of our experiments were guided by us considering our task as a machine translation task, if only for a lack of alternative approaches. We agree that from the point of view as an NMT, our task is somewhat peculiar, which is why we explicitly discuss those peculiarities in detail in the paper.
>
> > a lot of training is necessary to prime the model to learn this identity transformation, and that data is not given to those models.
>
> I am not entirely sure what is meant by "that data". It would have been possible (and we considered this) to pretrain NMT models to learn the identity on plain LaTeX fragment first, but we assumed that to "unlearn" the identity afterwards on symbolic expressions would be no less difficult for a such initialized model than to train a randomly initialized model in the first place - especially since learning the identity does not in fact teach the model anything with respect to the semantics of the sentences, which is what it needs to learn for correctly disambiguating from document context.
>
> > if automated methods produce the dataset, which is then used to train the model, then why are those methods not sufficient for the end task?
>
> We produce two parts of the dataset via automated means: We generate plain LaTeX from existing sTeX documents, and we synthesize training data by generating random sTeX and then translating that to plain LaTeX. In both cases, the easy step is to translate sTeX to plain LaTeX, which can be easily done by deterministic algorithms (it amounts to just expanding sTeX macros). The hard part that our paper addresses is the reverse: Generating (correct) sTeX from plain LaTeX, which requires some amount of document comprehension.
>
> > It was also bizarre in the results section how the baselines were dismissed in writing, their results were never presented.
>
> The results were entirely nonsensical due to the lack of training data. As mentioned, our experiments with established NMT models did not even produce syntactically valid LaTeX, so presenting these would not have been informative (examples include e.g. sequences of closing braces, or ungrammatical concatenated substrings of the training data with no relation to the current input).
>
> > If the baselines are truly that bad, then do they suffice as baselines?
>
> We agree that they are not meaningful baselines, but they are the only thing we could compare our model to at all. We would happily compare our model to autoformalization projects instead, once we have a compatible evaluation dataset, but as mentioned there is a large amount of work required in order to do so, for which we want to significantly simplify the data generation workflow first.
>
> > These are all things that are outside the scope of the typical ICLR paper and thus warrant a clear introduction, but space is limited.
>
> In deed, the amount of introduction required is a problem given the page limit. Since in the final version we would have an additional page of space, suggestions on which parts of the paper should be expanded most would be very welcome.

---

### Official Review · AnonReviewer1 · 2020-11-02
**An interesting work for autoformalization**

**Rating:** 6
**Confidence:** 3

**Review:**

This paper proposes a new task, disambiguating an informal math expression in LATEX by associating its tokens with concepts in a predefined formal math library and determining its abstract syntax tree. As argued in the paper, I agree that this task could serve as an important step for autoformalization, which is one of the most important problems of formal reasoning.

The task setup is reasonable. LATEX is commonly acceptable to be the informal language for editing math expressions. STEX and SMGLoM are powerful tools to annotate LATEX expressions with formal concepts. By advancing on this problem, we can greatly reduce the workload of autoformalization.

The drawback of the current benchmark is the lack of training and evaluation data. I think the lack of training corpora may be addressed by pretraining and building synthetic data. We do need a larger and high-quality evaluation set to validate any actual progress on this problem. The current evaluation set is too small and covers limited math topics. Also, the evaluation protocol is quite unclear. From what I understand, our best evaluation protocol should be checking if S_F belongs to STEX(S_STEX), which is not used in this work. Is there a way to implement this protocol?

The proposed approach looks fine. It is better to have an ablation study on the corresponding contributions of pretraining and synthetic data.

In general, I think this paper proposes an important task. By building a larger evaluation set and figuring out a clear evaluation protocol, this could be an important benchmark for the AI/TP community.

=======================================================================
After reading other reviews and authors' responses, I upgrade my score to 6. Despite its relatively small evaluation data, I think the setup of the task of autoformalization could still contribute to the community and inspire more researchers to make efforts in this direction.

---

> ### Author Response · Authors · 2020-11-16
> **Our reply**
>
> > We do need a larger and high-quality evaluation set to validate any actual progress on this problem.
>
> We fully agree. A larger evaluation set using the standards applied in the paper is (currently) a significant amount of work though, and resources are unfortunately very limited - especially now that the funding period for this project is over. Our criteria for the evaluation set were:
> 1. Unlike the plain LaTeX side of the training set, it should be entirely written by hand to avoid bias,
> 2. (Most, but ideally) all symbols occurring in the evaluation set should be aligned with a strongly typed library in order to allow for synthesizing training data (which we otherwise lack for most mathematical domains), and
> 3. The evaluation set should contain multiple symbols with the same presentation, so that non-trivial disambiguation becomes relevant - in our case primarily arithmetic operations on different domain sets (naturals, integers, reals, etc.).
>
> Extending the evaluation set, in particular to cover more mathematical topics, would be extremely desirable, but requires a lot of work and a non-trivial amount of expertise both regarding sTeX and the SMGloM as well as MMT and its formal libraries.
> We have ideas and are actively working on reducing the amount of effort involved in this though, so we hope that this will become more feasible in the not too distant future.
>
> > From what I understand, our best evaluation protocol should be checking if S_F belongs to STEX(S_STEX), which is not used in this work. Is there a way to implement this protocol?
>
> The problem here is that STEX(S_STEX) would be the set of all sTeX fragments that are correct full disambiguations of S_STEX (which is by definition already correctly fully disambiguated) - in other words, it is the set of all semantically equivalent sTeX-disambiguations. This set is necessarily not computable, so it's unclear how we could check that directly. However, the protocols we *did* implement are intended to be reasonable approximations of this, primarily "provided_stex" (i.e. string equality of S_F and S_STEX), "stexcheck" (i.e. S_F is fully disambiguated) and "stex_as_omdoc" (i.e. equality of syntax trees after translation to a strongly typed setting), the first and third of which do actually imply S_F\in STEX(S_STEX).

---

### Official Review · AnonReviewer4 · 2020-11-02
**Important step in autoformalization bringing in good tools**

**Rating:** 8
**Confidence:** 4

**Review:**

The paper presents a dataset for autoformalization (semantic
disambiguation) of informal Latex STEM documents. It is based on the
considerable amount of work that has been done in the last decade on
flexiformal (semi-formal) language formats and tools such as OMDoc,
OpenMath, sTeX and LaTeXML. The SMGloM glossary and the MiKoMH
repository are used as parallel sources, and the MMT system connecting
a number of formal systems and foundations is used for data
augmentation.

These are still relatively small datasets, so custom pretraining of
GPT-2 is done on the full arxiv corpus. The pretrained model is then
fine-tuned on the smaller training data.  Multiple evaluation metrics
that are meaningful in the semantic setting are defined - some of them
similar to those used in Wang et al 18 and Wang et al 20.

The final success rate of 47.2% of test data predicted correctly looks
very good and is comparable with the results of Wang18/Wang20 on the
synthetic data obtained by informalizing Mizar.

My overall impression is that this is an important step in the
autoformalization program [1]. It has involved a lot of work and brought
in a range of important tools developed recently.


Some detailed remarks:

p5: Disamiguation ==> spell check

p5: def 4.1 "We call S ∈ L fully disambiguated"
==>
I would not call the text fully disambiguated without types of
variables. In systems with subtypes (e.g., Mizar, possibly also other
PAs with typeclasses) the meaning and provability of a statement
(e.g., "forall x exists y st x = y *_complex y") will change depending
on whether the quantification is over complex, real, rational, integer
or natural numbers.


p7: Question: S_F = S_sTEX means exact string equality or after white space normalization, etc? If so can you say what are exactly the normalizations and what is the success rate before and after them?

- Would larger GPT models help?

- Would unsupervised learning like in Wang 20 be useful in some context here? The unsupervised methods seem to have improved a lot recently.

References:

[1] Cezary Kaliszyk, Josef Urban, Jirí Vyskocil, Herman Geuvers:
Developing Corpus-Based Translation Methods between Informal and Formal Mathematics: Project Description. CICM 2014: 435-439

---

> ### Author Response · Authors · 2020-11-16
> **Our reply**
>
> > Question: S_F = S_sTEX means exact string equality or after white space normalization, etc? If so can you say what are exactly the normalizations and what is the success rate before and after them?
>
> It is equality *after* normalization. Since normalization was applied to the training data, and is applied to a document *before* being input into the algorithm, the output can be expected to be (close to) normalized anyway. Therefore we don't consider it informative to evaluate the algorithm without normalization.
>
> The precise normalization can be found in latex/src/main/scala/com/fifom/latex/Normalize.scala (definition of "val cleanups"). "Remove" removes macros entirely (e.g. "Remove(semicolon)" removes \; macros), "Modifier" replaces macros with other tokens (e.g. Modifier(mid,...) replaces "\mid" by a simple "|" character token). Whitespaces get reduced to a single space token during parsing automatically (as TeX would do it).
>
> > Would larger GPT models help?
>
> the current experiment was performed on a relatively small set of sTeX symbols, as occuring in the (relatively short) evaluation dataset. For that, the size of the model probably does not make much of a difference. Once we scale up the (mathematical) domain of the model, it seems plausible that the size of the model would have a more pronounced impact.
>
> > Would unsupervised learning like in Wang 20 be useful in some context here?
>
> My impression is that unsupervised methods (by and large) require more training data than supervised ones. So likely yes, assuming we can get more sTeX data (with a broader range of sTeX symbols) in the future. Relying on synthesized data alone for unsupervised methods seems unwise to me.

---

### Decision · Program_Chairs · 2021-01-07
**Final Decision**

**Decision:**

Accept (Poster)

**Comment:**


This paper tackles the task of translating informal LaTeX
math into a formal representation annotated with abstract concepts (sTeX /
SMGloM).  The authors build a synthetic training data generation mechanism,
and construct an evaluation dataset by hand. The problem is tackled as machine
translation, and vanilla systems fail, while GPT-2 pretrained on LaTeX documents
performs well. The reviewers recognize the importance of this work, in an area
where data is not plentiful and benchmarking is difficult.  The authors do a
good job in presenting a difficult topic rather clearly, but I would encourage
the authors to continue improving the presentation, possibly with clearer examples
or figures.  The particular "copying bias" useful in this task, pointed out by a
reviewer, is indeed interesting and I encourage the authors to consider that
discussion and the thoughtful reviews deeply. Overall, this is a significant
contribution to the field and I recommend acceptance.